# Metabolic Studies in Organoids: Current Applications, Opportunities and Challenges

Elena Richiardone †, Valentin Van den Bossche † and Cyril Corbet *

Pole of Pharmacology and Therapeutics (FATH), Institut de Recherche Expérimentale et Clinique (IREC), UCLouvain, Avenue Hippocrate 57, B1.57.04, B-1200 Brussels, Belgium; elena.richiardone@uclouvain.be (E.R.); valentin.vandenbossche@uclouvain.be (V.V.d.B.)
* Correspondence: cyril.corbet@uclouvain.be
† These authors contributed equally to this work.

**Abstract:** Organoid technologies represent a major breakthrough in biomedical research since they offer increasingly sophisticated models for studying biological mechanisms supporting human development and disease. Organoids are three-dimensional (3D) physiological in vitro systems that recapitulate the genetic, histological and functional features of the in vivo tissues of origin more accurately than classical cell culture methods. In the last decade, organoids have been derived from various healthy and diseased tissues and used for a wide range of applications in basic and translational research, including (cancer) tissue biology, development, regeneration, disease modeling, precision medicine, gene editing, biobanking and drug screening. Here, we report the current applications of organoid models to study (stem) cell metabolism in several pathophysiological contexts such as cancer and metabolic diseases. More precisely, we discuss the relevance and limitations of these 3D cultures to model and study metabolic (dys)functions associated with hepatic, renal or pancreatic disorders, as well as tumor development and progression. We also describe the use of organoids to understand the dynamic interaction between diet, microbiota and the intestinal epithelium. Finally, this review explores recent methodological improvements in organoid culture that may help to better integrate the influence of microenvironmental conditions in the study of tumor cell metabolic phenotypes.

**Keywords:** metabolism; organoids; metabolic diseases; stem cells; tumor microenvironment; personalized medicine

## 1. Introduction

Organoids are in vitro 3D structures in which cells spontaneously self-organize into progenitors and differentiated functional cell types, recapitulating complex aspects of the tissue of origin, from physiological processes to regeneration and disease [1]. They are primarily generated from primary tissues (single cells or tissue chunks) or stem cells such as adult stem cells (ASCs), induced pluripotent stem cells (iPSCs), or embryonic stem cells (ESCs). Since the seminal work of Sato and colleagues more than ten years ago [2] describing the establishment and growth of self-renewing intestinal epithelia that model the crypt–villus architecture, an extensive body of work has improved the methodology and expanded the range of tissues that can be studied [3,4]. Currently, ASC-derived organoids are established via two main methods, namely submerged culture (Figure 1a), which typically solely involves epithelial cells, and air–liquid interface (ALI) culture (Figure 1b), a more organotypic method that includes epithelial cells alongside integrated stromal and immune cells. In the former, organoids are embedded within solid gels of extracellular matrix (e.g., laminin-rich Matrigel, basement membrane extract (BME)) submerged beneath culture medium containing a tissue-specific combination of growth factors (e.g., ligands from the Wnt pathway, such as Wnt3a and/or R-spondin, epidermal and fibroblast growth factors (EGF, FGF) and the bone morphogenetic protein (BMP) inhibitor noggin) to allow

ASCs to undergo long-term self-renewal and differentiation into all cell lineages. Moreover, Rho-associated protein kinase (ROCK) inhibitors have been shown to greatly increase the efficiency of organoid generation since they prevent the activation of programmed cell death, anoikis, as well as stress and injury responses upon cellular disaggregation during tissue processing. By comparison, the ALI method allows organoid initiation and long-term culturing upon the plating of mechanically dissociated tissue fragments into a type I collagen matrix on top of a permeable filter, with direct air exposure, thereby facilitating oxygen diffusion [5–7]. Such methodology has shown the ability to grow large multicellular organoids by preserving native tissue architecture, such as the epithelium, "en bloc" with endogenous immune and stromal elements, without reconstitution (i.e., indirect co-culture). Notably, ALI-based organoid cultures do not require the supplementation of exogenous growth factors, probably thanks to the production of essential endogenous niche factors by stromal cells. Finally, organoids can be derived from iPSCs through a series of differentiation steps, by culturing them with specific growth and signaling factors, which results in the generation of the desired tissue type (Figure 1c).

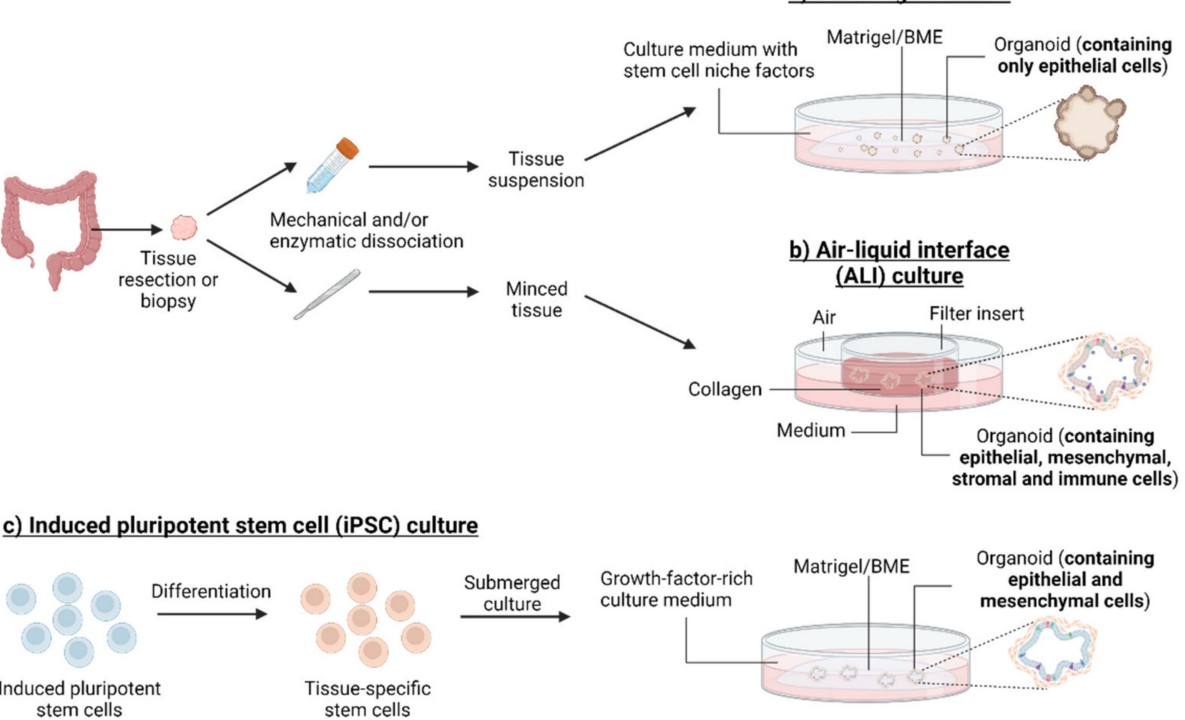

**Figure 1. Methods for organoid culturing**. (**a**) **Submerged culture**: upon harvesting and mechanical/enzymatic dissociation of tissues to obtain a single cell suspension, ASCs are embedded into a laminin-rich extracellular matrix (e.g., Matrigel or BME) and overlaid with a culture medium containing stemness-promoting factors to generate epithelial organoids. (**b**) **Air–liquid interface (ALI) culture**: alternatively, minced adult tissues can be cultivated in a type I collagen matrix within an inner Transwell dish. The top of the collagen gel is exposed to air, allowing a sufficient oxygen supply for cells. ALI method enables the generation of organoids comprising epithelial cells with their surrounding stroma, including fibroblasts and immune cells. (**c**) **Induced pluripotent stem cell (iPSC) culture**: iPSC-derived organoids are generated by a stepwise differentiation protocol resulting in the generation of the desired tissue type, often with the accompanying stroma. iPSC-derived organoids are then initiated by using submerged culture.

Advancements in 3D culture approaches have enabled organoids to be used to study various genomic, transcriptomic and proteomic alterations in many different pathophysiological situations. Although metabolic dysfunctions are now undoubtedly associated with

a variety of diseases, studies describing the use of organoids to decipher cell metabolism are still scarce. In this review, we report the current applications and limitations of organoid cultures to model and study cell metabolism in metabolic diseases and cancer. We also describe the use of organoids to understand the dynamic interaction between circadian rhythms, diet, microbiota and the intestinal epithelium. Finally, this review explores recent methodological improvements that may help to better integrate the influence of microenvironmental conditions in the study of tumor cell metabolic phenotypes.

## 2. Organoid Models to Study the Metabolic Control of Stem Cell Function

Stem cells have the capacity for self-renewal and multipotent differentiation, which enables the regulation of tissue development and homeostasis. Several studies have indicated a role for metabolism in the function of some adult stem cell populations [8–10]. Based on complex microenvironmental cues, stem cells can indeed modify their metabolic preferences, in particular switching between glycolysis and mitochondrial oxidative metabolism, to fulfil bioenergetic and/or biosynthetic requirements during lineage specification, differentiation or maintenance [11]. The intestinal crypt is a perfect example to illustrate the influence of a microenvironmental niche on stem cell metabolic preferences and tissue homeostasis. Okkelman and colleagues have reported the application of live cell microscopy of oxygen, via the phosphorescence lifetime imaging microscopy (PLIM) method, to provide high-resolution real-time visualization of oxygen distribution in mouse intestinal organoids [12,13]. More precisely, the use of a cell-penetrating phosphorescent $O_2$-sensitive probe revealed high heterogeneity in organoid oxygenation, with the existence of $O_2$ microgradients, depending on the age of the culture and drug treatment, thereby indicating that integrating the metabolic heterogeneity is critical for proper data interpretation with organoids. In two other studies, the same authors also used fluorescence lifetime imaging microscopy (FLIM) to assess NAD(P)H levels as well as mitochondrial membrane potential and obtained a quantitative, multi-parameter, live readout of the balance between cell redox and energy production within the intestinal stem cell (ISC) niche [14,15] (Figure 2a). Other studies have allowed the measure of the real-time bioenergetic profile (i.e., oxygen consumption and extracellular acidification rates) of intestinal crypt organoids [16–18]. For instance, by using small intestinal organoids that recapitulate crypt structure and the interaction between Paneth cells and Lgr5+ (leucine-rich repeat-containing G protein-coupled receptor 5-positive) stem cells (i.e., crypt base columnar cells (CBCs)), Rodriguez-Colman and colleagues elegantly identified a role of metabolism in the maintenance of stem cell function [19]. The authors have described metabolic compartmentalization within organoids, with CBCs displaying high mitochondrial activity (as reflected by a high pyruvate/lactate ratio and increased mitochondrial membrane potential) while adjacent Paneth cells exhibit an enhanced glycolytic metabolism. Importantly, they have documented the establishment of a lactate-based metabolic symbiosis between the two cell types, with the lactate produced by Paneth cells serving as a respiratory substrate to sustain oxidative metabolism in Lgr5+ CBCs (Figure 2a). The interruption of this lactate shuttle with specific metabolic inhibitors in either cell type leads to decreased organoid reconstitution, thereby supporting the existence of a metabolic niche that provides optimal stem cell function in the intestinal crypt [19]. In proliferating Lgr5+ stem cells located at the base of the intestinal crypt, the inhibition of mitochondrial pyruvate import upon genetic deletion of *MPC1* has been found to increase proliferation and expand the stem cell compartment [20]. A similar observation was made in intestinal organoids upon pharmacological inhibition of the mitochondrial pyruvate carrier (MPC) with UK-5099. In another study, jejunum-derived mouse organoids (enteroids) were used to assess the role of glutamine metabolism on ISC function [21]. The authors have shown that glutamine (as well as L-alanyl-L-glutamine, a stable glutamine dipeptide) supports intestinal epithelial homeostasis by promoting stem cell expansion, mTOR signal activation and crypt regeneration. Glutamine-deprived enteroids display gradual atrophy of crypt-like domains, with decreased epithelial proliferation via the activation of a reversible quiescent state in ISCs while maintaining Paneth and goblet cell

differentiation. By using a similar model of jejunum-derived mouse enteroids, another study confirmed the important role of amino acid metabolism in the self-renewal and differentiation potential of ISCs [22]. The authors have indeed found that deprivation of the essential amino acid methionine markedly reduces the proportion of Lgr5+ stem cells and suppresses cell proliferation in enteroids, while enhancing expression of the enteroendocrine cell marker chromogranin A as well as markers of enterochromaffin, goblet and Paneth cells.

## a) Metabolic heterogeneity within intestinal organoids

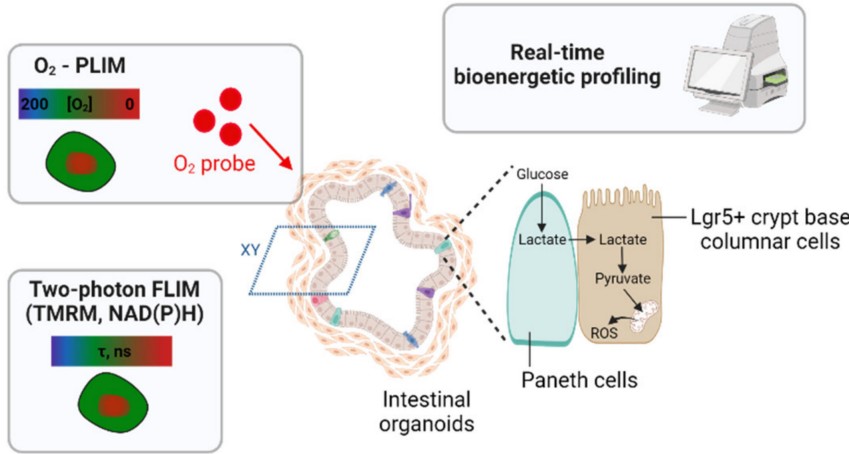

## b) Metabolic dynamics during kidney organoid differentiation

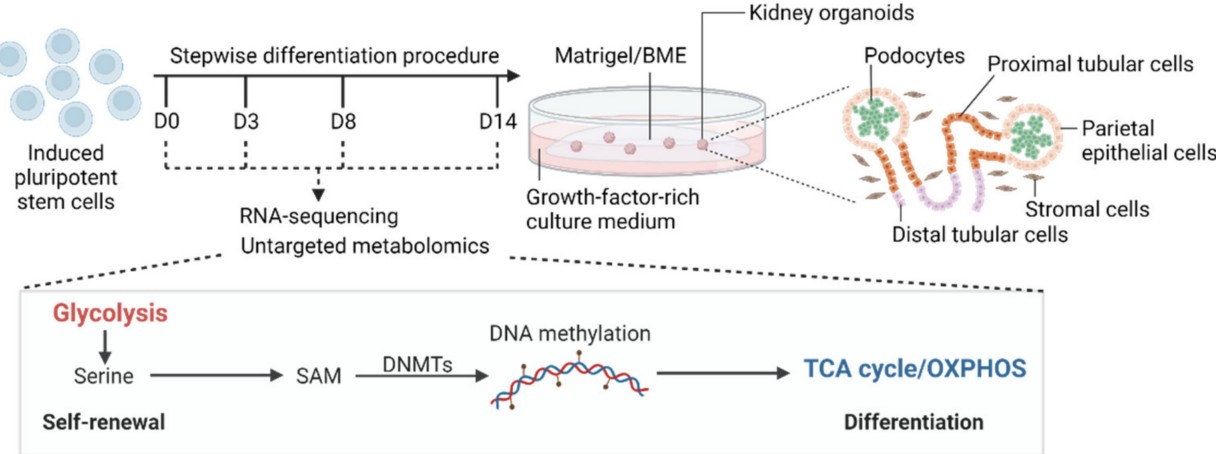

**Figure 2. Organoids for studying metabolic control of stem cell function.** (**a**) Several methodological approaches, including phosphorescence (PLIM) and fluorescence (FLIM) lifetime imaging and real-time bioenergetic profiling, have revealed a metabolic heterogeneity within intestinal organoids, in particular, with the existence of a lactate-based metabolic symbiosis between Paneth cells and Lgr5+ crypt base columnar cells. (**b**) Metabolic characterization of iPSC-derived kidney organoids has shown a metabolic switch from glycolysis to mitochondrial OXPHOS and tricarboxylic acid (TCA) cycle and the specific role of serine metabolism during kidney organoid differentiation. DNMT: DNA methyltransferase, ROS: reactive oxygen species, SAM: S-adenosylmethionine, TMRM: tetramethylrhodamine methyl ester.

Several studies using intestinal organoids have also revealed a major contribution of lipid metabolism in the regulation of ISC activity. Indeed, high-fat or cholesterol-rich diets have been reported to alter the balance between self-renewal and differentiation in ISC and to endow organoid-initiating capacity to progenitors [23,24]. Fatty acid oxidation

(FAO) has been found to support ISC renewal upon several dietary changes, such as short-term fasting [25] and high-fat diet [26], with hepatocyte nuclear factor 4 (HNF4) and peroxisome proliferator-activated receptor (PPAR) transcription factors as key regulators of the expression of FAO-related genes [26,27]. Pharmacological inhibition of carnitine palmitoyltransferase 1A (CPT1A), the mitochondrial acyl-CoA transporter, with etomoxir, reduced the diet-induced crypt organoid-forming capacities [25,26]. Importantly, since most intestinal tumors originate from ISCs [28–30], these studies have also shown that a high-fat diet, as well as excess dietary cholesterol, can promote intestinal tumor initiation, with this effect being delayed upon FAO inhibition [23,24,26,31]. Along the same lines, Sebastian and colleagues have elegantly shown, in a recent study, the presence of metabolic heterogeneity among intestinal epithelial cells by using adenoma-derived intestinal organoid models [32]. They reported that enhanced glycolytic metabolism, in the absence of SIRT6 deacetylase, drives intestinal tumorigenesis by increasing the number and activity of ISCs and by promoting their tumor-initiating potential. The authors have also identified a specific subpopulation of quiescent cells, with high pyruvate dehydrogenase kinase activity and increased stem cell potential, thereby providing new insights into the role of metabolism in ISC activity regulation and intestinal tumorigenesis.

Besides their applications in the understanding of ISC function, organoid models have been exploited to decipher stem cell metabolism in other tissues. Human pluripotent stem cell-derived cardiac organoids (hCOs) have been used to identify central regulators of the maturation process in the mammalian heart during postnatal life [33]. The use of hCOs in a miniaturized semiautomated cardiac organoid culture platform revealed that simulating the postnatal metabolic switch towards fatty acid utilization (at the detriment of carbohydrates) induced an increased expression of adult sarcomeric protein isoforms and cell cycle arrest. More precisely, the authors have found that a low-carbohydrate, low-insulin, palmitate-based medium represses key proliferation pathways, including β-catenin and Yes-associated protein 1 (YAP1) signaling, while inducing DNA damage response in hCOs, thereby supporting cardiac maturation. In another study, kidney organoids derived from iPSCs have been metabolically characterized to investigate the metabolic dynamics and function during kidney organoid differentiation [34]. Transcriptomics and untargeted metabolomics analyses validated a metabolic switch from glycolysis to mitochondrial oxidative phosphorylation (OXPHOS) during the iPSC differentiation process, also revealing a role for glycine, serine and threonine metabolism in the regulation of kidney organoid formation and lineage maturation (Figure 2b). More precisely, the authors described the contribution of serine metabolism in kidney organoid differentiation by regulating the production of S-adenosylmethionine and altering subsequent DNA methylation processes. Human pluripotent stem cell (hPSC)-derived retinal organoids have also allowed the characterization of metabolic changes accompanying photoreceptor differentiation [35]. By applying FLIM and hyperspectral imaging of organoids, the authors have revealed spatial and temporal metabolic changes during retinal organoid maturation, with increased glycolytic activity (detected with a higher free/bound NADH ratio) as well as retinol and retinoic acid accumulation in the organoid outer layer, coinciding with photoreceptor genesis. Finally, iPSC-derived human hepatic organoids have been generated and thoroughly characterized at the histological, transcriptional, metabolic and functional levels [36]. While the metabolic reprogramming of iPSCs from glycolysis to mitochondrial OXPHOS has been shown during organoid generation, transcriptomic analysis also revealed the increased expression of genes encoding proteins involved in many pathways of lipid metabolism, including the uptake and oxidation of exogenous fatty acids as well as cholesterol metabolism, upon differentiation. Altogether, these observations highlight the great potential of organoid models to reveal important roles of spatially and temporally controlled metabolic heterogeneity in the stem cell function and tissue homeostasis.

## 3. Organoids and Metabolic Diseases

Research in organoid models has opened up endless new possibilities for the study of metabolic diseases, including metabolic (dys)functions associated with hepatic, renal or pancreatic disorders, through the development of new models to assess functionality, pathogenicity and response to treatments [37,38].

### 3.1. Organoids to Model Liver Metabolic Diseases

Liver organoids, derived from either adult tissues (e.g., surgical resection, liver transplantation, needle biopsy) or iPSCs, are composed of multiple hepatic cell types, including hepatocytes, hepatic stellate cells and Kupffer cells. Several culture protocols have been reported to establish 3D liver organoids for the study of hepatocyte function, including glucose metabolism under normal and stress conditions [39], drug metabolism [40], as well as several hepatic metabolic diseases, including alpha-1 antitrypsin (A1AT) deficiency, citrullinemia type-1 (CTLN1), Wilson's disease, Wolman's disease and non-alcoholic fatty liver disease (NAFLD) [41–43] (Table 1).

**Table 1.** Organoid-based models of kidney and liver metabolic diseases.

| Disease | Species | Organoid Source and Derivation | Refs |
|---------|---------|--------------------------------|------|
| **Kidney metabolic diseases** | | | |
| Fabry disease | Human | iPSCs (fibroblast-derived) | [44] |
| **Liver metabolic diseases** | | | |
| Alpha-1 antitrypsin deficiency | Human | Adult tissue (surgical resection; liver transplantation; biopsy) | [45,46] |
| Citrullinemia type 1 | Human | iPSCs (fibroblast-derived) | [47] |
| Steatosis, steatohepatitis | Human | iPSCs (fibroblast-derived) | [48] |
| | Cat | Adult tissue (post-mortem) | [49,50] |
| Wilson's disease | Dog | Adult tissue (surgical resection, needle biopsy, fine needle aspiration) | [51,52] |
| Wolman's disease | Human | iPSCs (fibroblast-derived) | [48] |

### 3.1.1. Alpha-1 Antitrypsin Deficiency and Citrullinemia Type-1

A1AT deficiency is an inherited metabolic disorder caused by mutations in the *SERPINA1* gene and characterized by low circulating levels of A1AT, a serine protease inhibitor known to protect the lung against proteolytic damage from neutrophil elastase. A1AT deficiency-related liver disease occurs due to the aberrant folding and subsequent intracellular retention of the mutant protein, thereby triggering endoplasmic reticulum (ER) stress and apoptosis in hepatocytes [53]. Huch and colleagues have reported that differentiation of cholangiocyte-derived liver organoids, established by using biopsies from patients with A1AT deficiency, could recapitulate in vitro fundamental characteristics of the disease, including the accumulation of A1AT protein aggregates, the reduction of A1AT secretion, the induction of ER stress and increased apoptosis [46]. Similar observations were recently reported when using liver organoids derived from patients with different A1AT deficiency-causing genotypes, thereby highlighting the potential of organoids for A1AT deficiency-related liver disease modeling [45].

Human iPSC-derived hepatic organoids have been used to model CTLN1, a urea cycle disorder caused by mutations in the gene encoding the argininosuccinate synthetase 1 (ASS1) enzyme that is essential for the conversion of excess ammonia into urea [47]. Indeed, the authors have shown that CTLN1 organoids exhibit an increased accumulation of ammonia in comparison to organoids derived from healthy controls while maintaining other important functions such as albumin secretion, glycogen storage and lipid uptake, as observed in patients. Importantly, they have reported that ammonia detoxification can

be rescued in liver organoids upon overexpression of the wild-type *ASS1* gene, thereby paving the way for gene therapy in CTLN1 patients.

### 3.1.2. Wilson's and Wolman's Diseases

Wilson's disease is a rare genetic disorder caused by pathogenic loss-of-function variants in the *ATP7B* gene encoding a copper-dependent ATPase. It is characterized by disrupted copper homeostasis resulting in liver disease and/or neuropsychiatric symptoms. In dogs, mutations in the copper metabolism domain-containing 1 (*COMMD1*) gene have been associated with a defective biliary excretion of copper and with hepatic disorders similar to the clinical features observed in human patients with Wilson's disease [54]. Nantasanti and colleagues have shown that canine liver organoids established from dogs with an autosomal recessive *COMMD1* deficiency exhibit a higher intracellular accumulation of copper, compared to normal organoids, similar to the copper excretion defect in the in vivo situation [52]. Importantly, re-expression of a functional wild-type *COMMD1* gene could restore copper excretion and improve liver organoid viability upon $CuCl_2$ treatment. In a follow-up study, the same authors have provided preclinical proof of concept for organoid-based cell transplantation in vivo [51]. They have documented the use of autologous gene-corrected liver organoids for cell transplantation in a canine *COMMD1*-deficient model of copper storage disease. Although their results have shown low engraftment and repopulation rates, they have documented organoid cell survival up to two years post-transplantation.

Wolman's disease is a severe form of lysosomal acid lipase (LAL) deficiency characterized by the accumulation of lipids in the tissues and organs of the body due to an impaired breakdown of triglycerides. Mutations in the LAL-encoding gene lead to a reduced or absent function of the enzyme, thereby triggering hepato(spleno)megaly and hepatic failure [55]. While the disease shares many characteristics of hepatic steatosis, life-threatening complications often develop during early childhood and it can be lethal in absence of treatment. Although Kanuma® (sebelipase alfa) is now approved as the first treatment for patients with LAL deficiency, new effective therapies are still urgently needed. By using iPSC-derived liver organoid models of steatohepatitis, Ouchi and colleagues confirmed a significant increase in lipid accumulation in organoids derived from patients with Wolman's disease compared to healthy controls, a phenotype rescued by ectopic re-expression of the LAL enzyme [48]. These models were also used to test the farnesoid X receptor (FXR) agonist, obeticholic acid, for its efficiency in suppressing lipid accumulation in diseased organoids, thereby offering new therapeutic opportunities for patients with Wolman's disease.

### 3.1.3. Non-Alcoholic Fatty Liver Disease

NAFLD encompasses a range of conditions caused by a build-up of fat in the liver and it ranges from hepatic steatosis to non-alcoholic steatohepatitis (NASH), a more serious form of the disease that includes liver inflammation and hepatocyte damage. NASH can progress to cirrhosis and liver failure as well as to the development of hepatocarcinoma. Current in vitro and in vivo models for NASH-related research, including primary human hepatocytes, hepatoma cell lines and animal models, have shown strong limitations in the identification and the study of new personalized therapies, thereby highlighting an urgent need for a predictive human model to assess the efficacy of anti-inflammatory and anti-fibrotic drugs. For this purpose, Ouchi and colleagues developed a multicellular iPSC-derived liver organoid model, composed of hepatocyte-, stellate- and Kupffer-like cells, which exhibit highly comparable transcriptomic signatures with the in vivo-derived tissues. Upon free fatty acid (FFA) treatment, organoids could recapitulate in vitro the key features of steatohepatitis, including steatosis, inflammation and fibrosis, in a sequential manner [48]. Remarkably, atomic force microscopy demonstrated that increased stiffness in organoids following FFA exposure was correlated with the severity of fibrosis. Similarly, another study reported lipid accumulation in liver organoids of human, mouse, cat and

dog origin following treatment with FFA [50]. In a follow-up study, the same authors used feline liver organoid models to test drugs for their potential to reduce lipid accumulation and they identified T863 and AICAR (diacylglycerol O-acyltransferase 1 inhibitor and adenosine monophosphate-activated protein kinase activator, respectively), as two promising candidates for further clinical evaluation [49]. All these studies highlight the potential of organoids to model liver metabolic diseases and to offer new perspectives in personalized medicine and drug discovery.

### 3.2. Organoids to Model Pancreatic and Renal Metabolic Diseases

Organoid technology provides an opportunity to accelerate research on the pathophysiology of pancreatic and renal tissues through the development of 3D models that closely recapitulate key functional features of the organs.

### 3.2.1. Diabetes

Diabetes mellitus, characterized by hyperglycemia, is a group of chronic metabolic diseases affecting around 425 million people worldwide. While type 1 diabetes (T1D) is an autoimmune disease that results in the loss of insulin-producing β cells, leading to insulin deficiency, type 2 diabetes (T2D) is characterized by insulin resistance that may be combined with relatively reduced insulin secretion and it represents the most common form in adults. The pancreatic islet comprises several types of endocrine cells, including insulin-secreting β cells (~60%), glucagon-releasing α cells (~30%), somatostatin-releasing δ cells (~10%), pancreatic polypeptide-secreting PP cells and ghrelin-secreting ε cells [56]. By secreting hormones, islets are essential for controlling glucose homeostasis and islet transplantation has shown promise for the clinical management of patients with insulin-dependent diabetes. However, diabetes remains one of the most challenging health concerns since available drugs treat the symptoms but do not cure the disease. Human pancreatic islet organoids have recently emerged as promising models for diabetes to study the mechanisms supporting islet-related diseases, to test the potency and toxicity of novel therapeutic options and to serve as a considerable source of biological material for autologous islet transplantation, as extensively reviewed elsewhere [57–60]. Recent studies have optimized protocols to generate viable and functional islet organoids upon co-culturing of dissociated islet cells with human amniotic epithelial cells (hAECs) [61,62]. The authors have demonstrated that the incorporation of hAECs into islet organoids markedly improves engraftment, viability and graft function in a mouse T1D model, thereby offering new perspectives for the development of cell-based curative therapies for T1D patients. By using functionally mature iPSC-derived human islet-like organoids, another study has identified the non-canonical Wnt4 signaling as a main driver of the metabolic maturation necessary for efficient glucose-stimulated insulin secretion and has reported the potential of these models to re-establish glucose homeostasis in a diabetic mouse model [63]. Even more strikingly, the authors showed that ex vivo stimulation with interferon-γ induced endogenous expression of the immune checkpoint protein programmed death-ligand 1 (PD-L1) and suppressed T cell activation and graft rejection, thereby enabling the restoration of glucose homeostasis up to 50 days after transplantation. Altogether, these studies illustrate the great potential (but also current limitations) of organoid technology for diabetes research and therapy.

### 3.2.2. Kidney Diseases

Human and mouse iPSC-derived kidney organoids have been generated using different protocols [64–67]. An optimized method, based on the separate generation of nephron and ureteric bud progenitors before mixing in culture with embryo-derived stromal cells, has been shown to produce reassembled organoids in which the branching morphogenesis and inherent architecture of the embryonic kidney, including the peripheral progenitor niche and connection between nephrons and collecting ducts, were better recapitulated [68]. The generation and use of kidney organoids for biomedical research

have been extensively reviewed by others in recent years [69–72]. Organoid cultures have successfully been applied to model several kidney diseases, including polycystic kidney disease [67,73,74], nephronophthisis-related ciliopathy [75], mucin 1 kidney disease [76] and podocytopathies [77,78]. A recent study also reported the use of human iPSC-derived kidney organoids combined with the CRISPR-Cas9 genome editing system to model Fabry disease, an X-linked lysosomal storage disease, upon the introduction of a mutation in the galactosidase alpha (GLA) gene [44]. The authors showed that GLA-mutant human kidney organoids phenocopied human Fabry nephropathy by inducing deformed podocytes and tubular cells with accumulation of globotriaosylceramide, similar to organoids expressing the wild-type enzyme. By using transmission electron microscopy and oil red O staining, they also documented the intracellular accumulation of lipid droplets, as well as damaged mitochondria, in kidney organoids expressing the mutant form of GLA. Gene expression profiling revealed a reduced metabolism of glutathione (GSH) in GLA-mutant kidney organoids and this was associated with increased oxidative stress and reactive oxygen species levels. Importantly, while enzyme replacement therapy with recombinant human α-Gal A attenuated oxidative stress as well as the structural and transcriptional changes in GLA-mutant human iPSC-derived kidney organoids, direct treatment with GSH could also reduce cell death and reverse the gene expression pattern (i.e., re-expression of podocyte and tubular markers), thereby positioning GSH as an efficient therapeutic option for Fabry disease. Finally, another recent study has described the establishment and use of hPSC-derived diabetic-like kidney organoids to unravel the molecular mechanisms underlying the higher susceptibility to SARS-CoV-2 infections in human diabetic patients [79]. The authors have observed that kidney organoids derived from diabetic patients display some metabolic alterations, including decreased mitochondrial biogenesis, increased oxidative metabolism and enhanced glycolysis, which result in increased susceptibility to SARS-CoV-2 infection. Indeed, the treatment of kidney organoids with dichloroacetate, an inhibitor of mitochondrial pyruvate dehydrogenase kinase, has been shown to increase mitochondrial oxidative metabolism (to the detriment of glycolysis) while reducing SARS-CoV-2 infection. These data provide new mechanistic insights to explain SARS-CoV-2 susceptibility in diabetic patients and open the door for the development and use of metabolism-interfering therapeutic strategies in the clinical management of COVID-19 patients.

## 4. Organoids for Modelling Diet–Microbiome–Host Interactions

Another field of application for organoid models is the study of the dynamic interaction between diet, microbiota and the host intestinal epithelium [80–82]. Complex regulatory networks and crosstalks occur between the host, its diet and its gut microbiota, and participate in the homeostasis of intestinal epithelial cells. Microorganisms and gut microbial metabolites have pivotal roles in a wide range of biological processes, including the regulation of lifespan, bioavailability and the biological activities of diet-, pharmaceuticals- and xenobiotics-derived compounds, thereby highlighting the need to extend our knowledge of diet–microbiome–host interactions. Intestinal organoids have rapidly proven to be valuable systems to overcome the limitations encountered with animal models or immortalized human cell lines grown as 2D monolayer cultures for such studies. They self-organize in vitro into 3D structures that closely recapitulate the tissue of origin [83] and they have been reported as relevant in vitro systems for concurrently studying nutrient transport, sensing and hormone secretion [84], drug uptake and metabolism [85], nutrient transport physiology during digestion [86], as well as dietary fat absorption [87]. By using intestinal organoids, Cai and colleagues have observed that, although several dietary constituents do not affect organoid growth, caffeic acid exhibits growth-inhibitory effects in a dose-dependent manner by reducing crypt-like structure formation [88]. However, contradictory effects were obtained with other compounds, such as monosodium glutamate and chlorogenic acid, thereby highlighting the need for future studies to carefully explore effects of phytochemicals on intestinal organoids. By using mouse gut organoids, Lukovac and colleagues documented that microbially produced short-chain fatty acids (SCFAs),

including acetate, propionate and butyrate, regulate distinct gene expression profiles [89]. More precisely, they showed that the exposure of mature ileal organoids to supernatants collected from *Akkermansia muciniphila* affected several transcription factors and genes involved in fatty acid, cholesterol and bile acid metabolism, such as *NR1H3*, *CPT1A* and *HMGCS1*. Another study using intestinal organoids also revealed that butyrate treatment could improve tight junction integrity in intestinal epithelial cells, decrease apoptosis and mitigate graft-versus-host disease [90]. Moreover, Schilderink and colleagues showed that the butyrate-induced upregulation of ALDH1A1 and ALDH1A3 in human and mouse intestinal organoids contributed to retinoic acid production and the maintenance of gut homeostasis [91]. By exploiting the intestinal organoid technology, Bellono and colleagues also demonstrated the role of serotonergic enterochromaffin cells as chemosensors in the gut epithelium to detect and transduce environmental, metabolic, and homeostatic information from the gut directly to the nervous system [92]. Microbiota-derived SCFAs were shown to improve metabolic activation in human iPSC-derived liver organoids by promoting CYP3A4 expression and albumin secretion and identified troglitazone-induced hepatotoxicity [93]. Importantly, the latter effect could be reversed upon treatment with ketoconazole, a potent CYP3A4 inhibitor, highlighting the great potential of this culture system to evaluate CYP3A4-dependent drug toxicity. Finally, a recent study by Rosselot and colleagues elegantly showed that human intestinal organoids possess circadian rhythms and exhibit circadian phase-dependent necrotic cell death responses to *Clostridium difficile* toxin B [94]. Importantly, the authors showed that the source utilized for organoid generation was critical since iPSC-derived intestinal organoids did not display circadian rhythms while more mature structures such human intestinal enteroids exhibited robust circadian oscillation of key clock-controlled genes, thereby providing new insights for the use of organoid models in circadian medicine [95].

Altogether, these studies demonstrate that intestinal organoids recapitulate the gut microenvironment and, therefore, represent suitable models for studying the diet–microbiome–host interactions as well as circadian rhythms. This research field is still evolving and benefits from recent technical improvements, including the use of disrupted organoids [96] or microinjections [97], to accurately mimic the gut architecture, luminal accessibility and tissue polarity.

## 5. Organoids and Tumor Metabolism

Tumors are complex and dynamic ecosystems in which the subclonal populations of cancer cells must adapt their metabolism to support disease progression [98]. Within the tumor microenvironment (TME), the limited access to oxygen and/or nutrients and the accumulation of protons in local compartments, together with the presence of tumor-associated stromal and immune cells generate metabolic heterogeneity [99,100]. Thus, according to their local TME, cancer (stem) cells can use a variety of substrates (e.g., glucose, amino acids, fatty acids) to fulfill their need in energy (ATP), biosynthetic precursors and reduced cofactors (NADH, NADPH) so that facilitating their survival, proliferation, metastasis and the development of resistance to anticancer therapies [101–104]. The TME-driven metabolic phenotype of cancer cells (and surrounding cells) evolves de facto with time and tumor development. The relevant pre-clinical models are therefore needed to explore the intimate relationship between TME and (cancer) cell metabolic preferences and to design novel strategies aiming to integrate and therapeutically exploit TME- and therapy-mediated metabolic addictions in cancer cells [105,106]. Major recent advances in 3D culture technology have led to the development of patient-derived tumor organoids (PDTO) for a variety of cancer types, including breast [107], bladder [108], colorectal [109], head and neck [110], liver [111], pancreatic [112], gastrointestinal [113], glioblastoma [114], retinoblastoma [115] and prostate cancers [116]. Importantly, they have been shown to recapitulate the histological and genetic features, as well as drug response, of the original tumor, thereby making them miniaturized in vitro avatars of patient tumors. Moreover, they retain the physicochemical characteristics of TME, such as hypoxic gradients [117–119],

making them suitable tools for studying intratumoral metabolic heterogeneity. In the last decade, optical metabolic imaging (OMI) has been applied to quantify the fluorescence intensity and lifetime of NAD(P)H and FAD and detect distinct metabolic cell states within tumor organoids. Several studies have indeed shown that such technique allowed the identification of metabolic heterogeneity and the prediction of therapeutic response in organoid models of breast [120–122], colorectal [123], gastroenteropancreatic neuroendocrine [124], head and neck [125], and pancreatic cancers [126]. Besides fluorescence-based OMI, matrix-assisted laser desorption/ionization mass spectrometry imaging (MALDI-MSI) technology has been used in several studies to assess the distribution of anticancer drugs, as well as their metabolites, in a variety of PDTO models [127–129]. These data pave the way for the use of OMI and MALDI-MSI in PDTO as high-throughput platforms to identify optimal therapies for individual patients and to integrate cancer cell metabolism in the current genomics-driven cancer paradigm for precision oncology [130]. In addition, although metabolomics and $^{13}$C tracer analysis have been established as state-of-the-art techniques to determine the concentration of metabolites and the activity of metabolic pathways, respectively, there are very few examples of such studies in 3D organoid models. The metabolome of (tumor) organoids has been captured by using NMR [131] and targeted [132] or non-targeted [133–135] liquid chromatography (LC)-MS-based profiling. Neef and colleagues have recently performed LC-qTOF-MS-based metabolic and lipidomic profiling on colorectal cancer organoids to reveal changes upon treatment with 5-fluorouracil [136]. They observed major alterations in levels of metabolites involved in purine and pyrimidine metabolism, in accordance with the mechanism of action of the drug, thereby providing the first basis of evidence for assessing drug-induced metabolic response in 3D organoid models. However, since the influence of background signals derived from the basal membrane extract used for organoid culturing on metabolomics data processing is still not fully appreciated, studies reporting metabolomics analyses on organoid models remain scarce.

In most of the metabolic studies reported so far, organoids are solely composed of epithelial cancer cells embedded in an extracellular matrix (e.g., Matrigel) and cultured in a specific medium containing a defined combination of stem cell niche factors. Nevertheless, these models have demonstrated a great potential to reveal and study the epithelial–mesenchymal plasticity (EMP) of cancer cells [137–139] and they have been already used to screen drugs that can reverse the epithelial–mesenchymal transition [140]. This is of particular interest since many studies have reported a link between metabolism and EMP in development and disease, including cancer [141,142]. Still, the lack of stromal and immune cell components is actually a major limitation since tumor cells can either cooperate (i.e., metabolic symbiosis) or compete with these non-cancerous cell populations during tumor development [143]. For instance, while PDTO models of colorectal cancer were shown to recapitulate many of the genetic and transcriptomic features of donor tumors as well as response to anticancer drugs, they failed to reveal the high complexity of metabolism-related molecular alterations, as observed in primary tumors and partly in patient-derived tumor xenograft models [144]. Recent studies have reported new modalities of PDTO generation, such as the ALI method, to preserve "en bloc" cancer cells with tumor stroma, including cancer-associated fibroblasts (CAFs) and even functional native immune cells (T and B cells, myeloid cells, macrophages and NK cells) [145]. Alternatively, direct co-cultures of tumor organoids with autologous CAFs [146–148] or peripheral blood lymphocytes [149–152] have been described and may represent useful tools to study the metabolic communication within TME and better recapitulate the intratumoral metabolic heterogeneity (Figure 3).

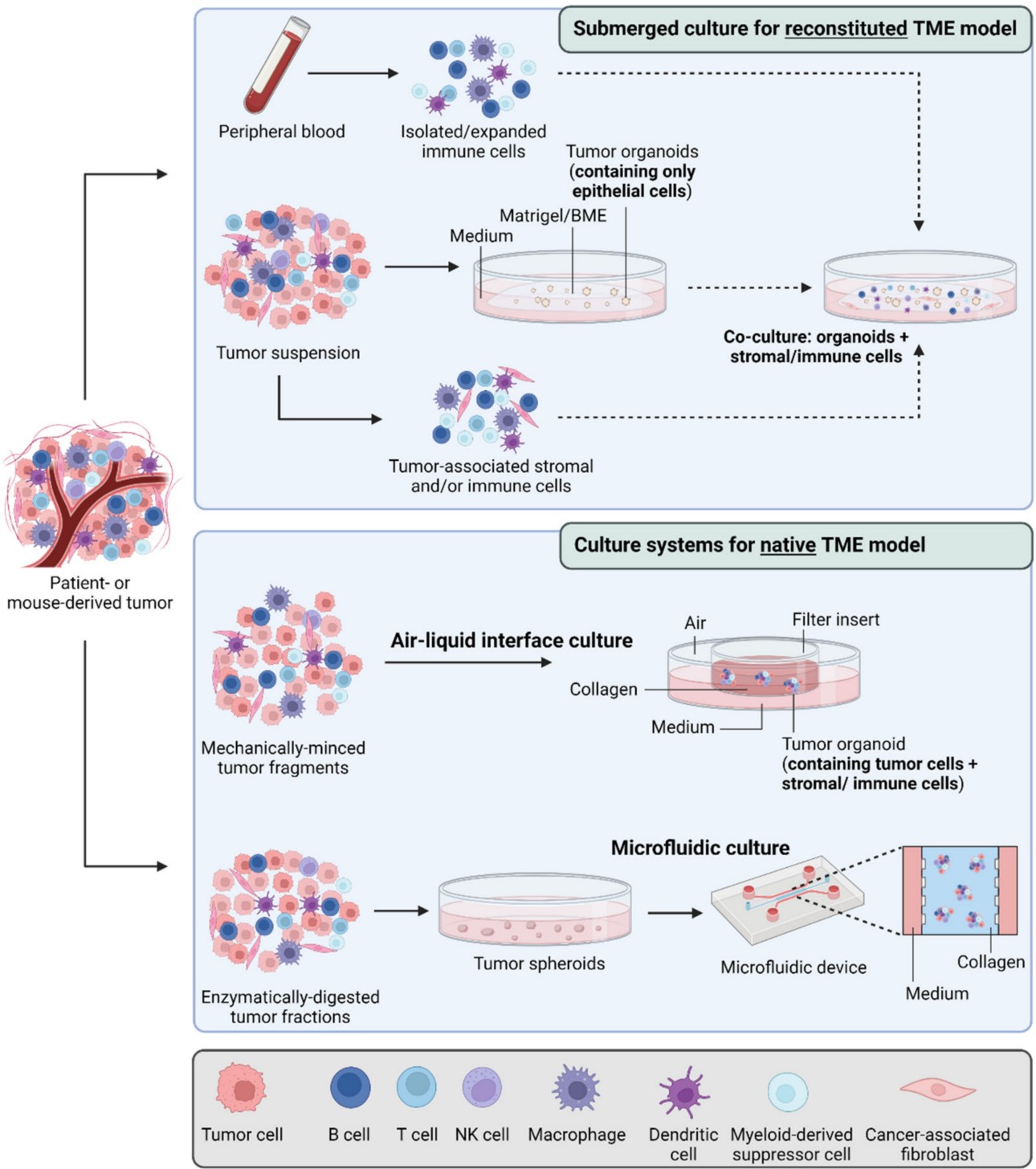

**Figure 3. Organoid culture methods to model TME conditions**. Stromal and immune cell components of the TME can be integrated within organoid cultures via two types of approach. The submerged culture enables the reconstitution of the TME upon direct co-culture of tumor epithelial organoids with stromal and/or immune cells obtained from autologous peripheral blood or tumor. Alternatively, the use of air–liquid interface culture or microfluidic culture devices allows the preservation of the native TME, including functional immune cells and cancer-associated fibroblasts (CAFs), from mechanically minced or enzymatically digested tumor fractions. NK cell: natural killer cell.

Recently, by using a co-culture model of pancreatic cancer organoids with pancreatic stellate cells (PSCs) combined with OMI technology, Datta and colleagues reported the

existence of a pyruvate-based metabolic symbiosis whereby fibroblasts facilitate oxidation reactions in cancer cells to support proliferation [153]. Another challenge in organoid cultures is the lack of functional vasculature, which may lead to the development of immature organoids and further prevent the study of anti-angiogenic drugs in these models. Co-culturing of human endothelial cells with mouse breast organoids has shown the development of a functional capillary vessel network, connected to the mice circulatory system, in an orthotopic model of human breast cancer [154]. Other studies have reported new experimental strategies to generate vascularized organoids [155,156], which may be further used for metabolic studies. Finally, although companies are developing synthetic scaffolds with the potential to offer xenogenic-free, chemically defined, highly tunable and reproducible alternatives [157], a major limitation of organoid cultures is still their reliance on an animal-derived extracellular matrix and defined medium conditions that might not represent nutrient levels encountered by cells within the tumor.

## 6. Conclusions

Organoid models are being rapidly integrated into various aspects of biomedical research and are constantly evolving due to improved derivation protocols and culture conditions. Applying organoids to the study of cell metabolism in a variety of pathophysiological contexts, such as metabolic diseases, cancer and diet–microbiome–host interactions, has the potential to greatly reduce the number of animal models used for equivalent purposes. Although the reduced cellular complexity in organoid models remains a major limitation for the study of metabolism in biological processes where multiple cell types interact in a spatial and highly dynamic way, recent improvements in culture methods and emerging technologies have the potential to allow metabolic characterization in organoids with single-cell and subcellular resolution (Table 2). We are thus convinced that all these new insights and novel technological approaches will help better assess metabolic (dys)function in a variety of diseases and will pave the way for the implementation of metabolic studies in organoids for disease detection, surveillance and treatment to improve outcomes and quality of life for patients with metabolic disorders.

**Table 2. Methods to study metabolism in organoids.** Here, the main methods reported to study cell metabolism in organoid models are listed. Their applications, advantages and disadvantages are indicated. ECAR: extracellular acidification rate; FAD: flavin adenine dinucleotide; NADH: nicotinamide adenine dinucleotide; OCR: oxygen consumption rate; TMRM: tetramethylrhodamine, methyl ester.

| Methods | Applications | Advantages | Disadvantages | Refs |
|---|---|---|---|---|
| Extracellular flux analysis (Seahorse XF analyzer) | Evaluation of mitochondrial respiration (OCR) and glycolysis (ECAR) | Real-time and simultaneous measurements<br>Up to 4 injections for nutritional and/or pharmacological modulation<br>Label-free assay system, highly sensitive microplate format | No spatial resolution<br>Technically challenging (e.g., need for accurate plating, optimal organoid density)<br>Use of saturating concentrations of substrates and drugs | [16–20] |
| Fluorescence lifetime imaging microscopy (FLIM)/Optical metabolic imaging (OMI) | Live cell microscopy of endogenous (NAD(P)H, FAD) or exogenous chromophores (TMRM for mitochondrial membrane potential) | Non-invasive, cell-specific and direct analysis of metabolism within organoid models<br>Compatible with other imaging methods (e.g., PLIM) for multiparametric quantitative analysis | Complex interpretation of fluorescence data (due to double or multi-exponential decays for most of fluorescent reporters)<br>Shorter lifetimes than PLIM | [14,15,120–126,153] |

**Table 2.** *Cont.*

| Methods | Applications | Advantages | Disadvantages | Refs |
|---|---|---|---|---|
| Mass spectrometry-based metabolomics | Absolute or relative quantification of extra- and/or intracellular metabolites within organoids | Suitable for targeted and untargeted profiling of several metabolite classes (e.g., lipids, polar metabolites) | No spatial resolution No information on metabolic flux (steady-state conditions) Background signal from animal-derived extracellular matrix | [133–135] |
| Phosphorescence lifetime imaging microscopy (PLIM) | Live cell microscopy of oxygen (with dedicated cell-penetrating phosphorescent $O_2$-sensitive probes) | Suitable for single-cell analysis of inter- and intra-organoid variability of oxygenation levels Direct, reversible and non-chemical process Compatible with other imaging methods (e.g., FLIM) for multiparametric quantitative analysis High sensitivity and high stability of the signal | Emission intensity dependent on the probe distribution within organoids Limited number of applications (depending on probe availability) | [12–14] |

**Author Contributions:** Conceptualization, C.C.; methodology; validation: E.R., V.V.d.B. and C.C.; writing—original draft preparation, E.R., V.V.d.B. and C.C.; writing—review and editing, E.R., V.V.d.B. and C.C.; supervision, C.C. All authors have read and agreed to the published version of the manuscript.

**Funding:** This research received no external funding.

**Acknowledgments:** All figures have been created with https://biorender.com/ (accessed on 9 June 2022).

**Conflicts of Interest:** The authors declare no conflict of interest.

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
