# Peer review of "Metabolic Studies in Organoids: Current Applications, Opportunities and Challenges"

_2674-1172, doi:10.3390/organoids1010008_

Round 1
Reviewer 1 Report
Richiardone et al. present here a comprehensive summary of the most relevant metabolism-related research done using organoids as an experimental system. In the last years, the development of organoid and 3D-culture technologies has greatly improved our understanding of fundamental biological processes in health and disease, including the role of metabolism in controlling cell fate, differentiation and transformation. In this regard, the authors have nicely discussed our current knowledge on the functional role of metabolism in these processes by focusing on those studies done with organoids. The manuscript is well written and organized, the figures are very clear and help to illustrate what is stated in the text. Many other reviews have been published on the topic of organoids, but this is one of the few focusing specifically on organoids as a model system to study metabolism and, therefore, I believe it would be of great interest for the scientific community. I only have few minor points that, in my opinion, could further improve an already excellent manuscript.
Minor concerns:
- As stated above, the manuscript is well structured and covers relevant biological processes, organoid models and metabolic techniques/experiments done in organoids. However, I think it would be nice to include a table with the most relevant techniques to study metabolism in organoids (metabolomics, metabolic imaging, metabolic reporters, extracellular flux analyses, etc), their advantages and their disadvantages. This would help the readers to have a snapshot of all current techniques and help them in deciding which experimental approach to follow for their research.
- In section 2, when describing the role of metabolism on ISCs, the authors could comment on the role of fatty acids, as there are several studies done by the Yilmaz's laboratory establishing a clear link between fatty acids and ISC activity.
- In section 3.2.2 (kidney diseases), the authors would benefit of including a very recent study showing that kidney organoids from diabetic patients exhibit altered mitochondrial respiration and enhanced glycolysis, resulting in high SARS-CoV-2 infection susceptibility (Garreta et al., Cell Metabolism 2022).
Author Response
As stated above, the manuscript is well structured and covers relevant biological processes, organoid models and metabolic techniques/experiments done in organoids. However, I think it would be nice to include a table with the most relevant techniques to study metabolism in organoids (metabolomics, metabolic imaging, metabolic reporters, extracellular flux analyses, etc), their advantages and their disadvantages. This would help the readers to have a snapshot of all current techniques and help them in deciding which experimental approach to follow for their research.
We would like to thank the reviewer for her/his valuable comments. We have now added a new Table 2 describing the main methods used to study metabolism in organoids. As suggested, we have indicated their advantages and disadvantages, as well as their main applications.
In section 2, when describing the role of metabolism on ISCs, the authors could comment on the role of fatty acids, as there are several studies done by the Yilmaz's laboratory establishing a clear link between fatty acids and ISC activity.
We thank the reviewer for her/his important suggestion. We have added a new paragraph about the role of lipid metabolism on ISC activity (lines 143-155). References 23-31 have been added.
In section 3.2.2 (kidney diseases), the authors would benefit of including a very recent study showing that kidney organoids from diabetic patients exhibit altered mitochondrial respiration and enhanced glycolysis, resulting in high SARS-CoV-2 infection susceptibility (Garreta et al., Cell Metabolism 2022).
As nicely suggested by the reviewer, we have now added this very recent study from Garreta et al in the section 3.2.2 (new ref 79).
Reviewer 2 Report
The review cover a very interesting field and has the value to be clear, with a high attention to the literatures available and strongly medical oriented.
I'd like to suggest just to take into in account into the paragraph concerning Organoids and tumor metabolism the role that Epithelial to mesenchymal plasticity may have in control metabolism.
Author Response
I'd like to suggest just to take into in account into the paragraph concerning Organoids and tumor metabolism the role that Epithelial to mesenchymal plasticity may have in control metabolism.
We would like to thank the reviewer for her/his valuable comments. In the "Organoids and tumor metabolism" section, we have now referred to the role of epithelial-to-mesenchymal plasticity on (tumor) cell metabolism and how/whether organoid models can be used to study this phenotypic feature. We have added several references in line with this concept:
-Zhang et al Am J Physiol Gastrointes Liver Physiol 2022
-Beerling et al Cell Rep 2016
-Hahn et al Sci Rep 2017
-Zhao et al Nat Commun 2021
-Jia et al Br J Cancer 2021
-Bhattacharya et al Dev Cell 2020